# Melatonin-Measurement Methods and the Factors Modifying the Results. A Systematic Review of the Literature

**DOI:** 10.3390/ijerph17061916

**Published:** 2020-03-15

**Authors:** Beata Rzepka-Migut, Justyna Paprocka

**Affiliations:** 1Department of Pediatric Neurology and Pediatrics, St. Queen Jadwiga’s Regional Clinical Hospital No 2 Rzeszów, 35-301 Rzeszów, Poland; beata-rzepka@o2.pl; 2Department of Pediatric Neurology, Faculty of Medical Sciences, Medical University of Silesia, 40-752 Katowice, Poland

**Keywords:** melatonin, measurement methods, body fluids, drugs, DLMO, Dim light melatonin onset

## Abstract

Melatonin plays an important role in regulating the sleep–wake cycle and adaptation to environmental changes. Concentration measurements in bioliquids such as serum/plasma, saliva and urine are widely used to assess peripheral rhythm. The aim of the study was to compare methods and conditions of determinations carried out with the identification of factors potentially affecting the measurements obtained. We have identified a group of modifiable and unmodifiable factors that facilitate data interpretation. Knowledge of modifiers allows you to carefully plan the test protocol and then compare the results. There is no one universal sampling standard, because the choice of method and biofluid depends on the purpose of the study and the research group.

## 1. Introduction

Melatonin (N-acetyl-methoxytryptamine; MLT) (Figure 1) is produced mainly by the pineal gland from tryptophan in the biochemical pathway and released according to the circadian rhythm dependent on the light–dark cycle. Stimulation of the pineal gland occurs in darkness whereas light suppresses its activity [1,2]. The pathway of transforming external light-related stimuli into the internal stimulus, triggering the production of MLT, occurs in a specific sequence of events. Light quanta are absorbed by the intrinsically photosensitive retinal ganglion cells (ipRGC) and transmitted through the retinohypothalamic pathway to the suprachiasmatic nucleus that is considered to be the central biological clock. Then, the signal goes to the paraventricular nucleus and through the upper thoracic intermediolateral cell column to the superior cervical ganglion, and then to the pineal gland through sympathetic fibers [1]. In the absence of a light stimulus, MLT production is regulated by feedback loops [3] and the biological clock has a periodicity of more than 24 h [4]. Melatonin is not stored and after its secretion it immediately diffuses into the blood and the cerebrospinal fluid (CSF) [5,6]. Melatonin concentration in the third ventricle is 20–30 times higher than in blood samples [5,7]. However, the further away from the pineal gland, the lower the MLT level in the CSF is [8]. Due to its lipophilic and hydrophilic properties, it diffuses easily through cell membranes and is detected in other body fluids, e.g., saliva, urine, milk, sperm and amniotic fluid. Different studies have also confirmed extra-pineal MLT synthesis in, e.g., the gastrointestinal tract, ovaries, lymphocytes, macrophages and retina. In the physiological rhythm, MLT concentration starts to increase between 9 and 10 p.m. The peak plasma concentration is reached between 3 and 4 a.m., decreases in the morning (between 7 and 9 a.m.) and has a low or undetectable concentration during the day [9]. Therefore, changes in plasma concentration are dynamic i.e., during the day its concentration is maintained at 5 pg/mL on average and it usually reaches 50–100 pg/ml at night [10].

About 70% of plasma MLT is bound to albumin and 30% is known as free MLT is excreted into saliva through passive diffusion. Hence, salivary MLT concentration accounts for 24%–33% of plasma MLT [11,12]. Melatonin is metabolized mainly in the liver to 6-hydroxymelatonin and then conjugated with sulfuric acid to 6-sulfatoxymelatonin (Figure 2) [9], which is the main MLT metabolite excreted in the urine. In addition, 3-hydroxymelatonin (Figure 3) and a small amount (about 1%) of unmetabolized MLT are also found in the urine [13]. The complex process of MLT secretion can be significantly impaired in certain neurological disorders [14,15,16].

## 2. Dim Light Melatonin Onset

An objective assessment of the circadian rhythm can be made by measuring parameters of physiological processes, the variability of which is largely regulated by the biological clock. Dim light melatonin onset (DLMO) is a widely recognized parameter indicating the time at which MLT levels begin to increase in the dark [17], which is used in the diagnosis and follow-up of the results of treatment of sleep disorders. It allows characterization of the circadian rhythm secretion of this hormone based on MLT concentration measurements only in blood, urine or saliva samples collected in dim ambient light [18]. Determination of DLMO is an alternative to actigraphy and measurements of deep body temperature. It is estimated that an increase in MLT concentration begins 2–3 h before bedtime [19] and its higher concentrations correlate with a decrease in vigilance, lower body temperature and decreased cognitive ability [3,13].

No uniform DLMO estimation protocol has yet been created. Studies differ in the sampling frequency, duration of observation and the adopted DLMO estimation threshold. Frequent sampling enables more accurate measurements, however it can be troublesome for the examined person and may affect the composition of the material (saliva) [20]. Crowley et al. estimated that sampling every hour as well as every 30 min is equally accurate [21]. Similar conclusions were presented by Molina et al. Additionally, they noted that hourly sampling is more practical, especially in large scientific studies [22]. DLMO assessment can be based on 24 h melatonin profiles or partial profiles with a sample window of usually 4–7 h. The combination of a partial melatonin profile with the patient’s sleep diary will increase the effectiveness of this method. The last ambiguous factor is the definition of the cut-off point. The universality of the definition used depends on the biological fluid in which the level of melatonin is determined. For blood, the most common definitions are the time when plasma melatonin levels increased to 25% of the matched night peak, based on the absolute threshold, the 10 pg/ml absolute threshold and the time point when melatonin levels were 2 SD above the baseline plus 15% of the 3 highest values. For saliva, scientists usually use an absolute threshold of 4 pg/ml, then 3 pg/ml and 2SD as the mean plus two SD of the three lowest consecutive daytime points. In a group of pediatric patients, the cut-off value is most often considered to be 4 pg/ml, and in groups of teenagers with similar frequency the threshold is 3 and 4 pg/ml. Crowley et al. 2016 proved the benefits of using a set absolute threshold, especially when we have a partial profile of melatonin [23] Although all definitions are accepted in the literature, the results obtained may vary significantly. Molina T et al. have estimated DLMO based on a fixed threshold of 3 pg/ml and a variable threshold of “3k”. They obtained time differences in DLMO by 22–24 min, regardless of sampling frequency [22] The diversity of the methodology adopted by researchers makes it difficult to compare the results of scientific publications.

## 3. Material Sampling

### 3.1. Blood

MLT has a short half-life ranging from 20–30 min [23] to 45–60 min [24]. Therefore, frequent sampling allows accurate current assessment of the synthesis of the hormone and the amount of MLT circulating at the time of material collection.

### 3.2. Urine

6-sulphatoxymelatonin-aMT6 (the main urine MLT metabolite) level is a good marker of plasma MLT level. Numerous studies show a close relationship between aMT6 concentration and nocturnal MLT level in plasma and peak nocturnal MLT [25,26,27,28]. To obtain strongly correlated data, the obtained aMT6 levels should be corrected for creatinine due to intraindividual variability [25,29]. As in the case of blood MLT concentrations, urinary aMT6 concentrations show a distinct circadian rhythm [30].

The correlation between measurements of aMT6 and serum MLT is more accurate when urine samples are obtained more frequently [12]. The phase shift between serum MLT level and urinary aMT6 was observed, which is related to metabolism [12]. This shift varies from 12 min to 2 h, depending on the adopted method [29,31]. These differences may result from MLT metabolism in the liver and subsequent renal clearance [25,32]. The level of aMT6 is also dependent on kidney function. Epidemiological studies on large samples i.e., 78 men [25] and 203 women [33] showed a significant relationship between nocturnal plasma MLT, morning urinary MLT and morning urinary aMT6 levels. The relationship between total nocturnal plasma MLT, urinary aMT6 referred to as creatinine and urinary MLT is significant.

### 3.3. Saliva

Salivary concentration of this hormone is about three times lower than plasma concentration. Studies demonstrate correlations between measurements of salivary MLT levels in respect to plasma [12] and urine aMT6 levels [34]. After administration of exogenous MLT, almost simultaneous changes were observed in salivary and serum MLT levels [35], which was also confirmed by another study [36]. Determining salivary MLT is a commonly used method due to the ease of sampling, relatively low invasiveness of the examination and reflection of the secretory profile of nocturnal MLT rhythm [34].

### 3.4. Cerebrospinal Fluid (CSF)

Examinations of the circadian MLT secretion conducted on human CSF samples are rare. Due to the fact that MLT is released directly into the CSF, data on its production and secretion are more accurate. Bumb et al. measured MLT levels in the CSF in young healthy volunteers before and after sleep deprivation and found higher morning levels after sleep deprivation (7.7 pg/mL vs. 3.2 pg/mL) [37]. Determination of MLT levels in the CSF is more frequently conducted using experimental animal studies [38].

## 4. Material Collection

At the time of material collection, standards should be uniform to minimize factors affecting the variability of determinations. Due to the circadian rhythm of MLT, samples should be obtained from the evening to morning hours. Measurements are more accurate when samples are collected more frequently. At the same time, subjects should be provided with a sufficient and undisturbed night’s rest. Otherwise, the results could be false. For this reason, a permanent intravenous or intravesical catheter could be used for blood and urine collection, respectively. However, there is no alternative that would allow sample collection without stimulating the subject during saliva sampling.

Patients should be in a dark room under low light intensity <50 lux [39]. Red dim light which does not interfere with MLT production is recommended [40]. 

Additional limitations are related to saliva collection. Subjects should neither drink nor brush their teeth for about 30 min prior to sample collection. Due to the possibility of false results, subjects should not use lipstick [41]. Chewing gum depends on the standard adopted. Some researchers did not stimulate saliva secretion in subjects. However, stimulation was used in some studies, e.g., paraffin chewing [42].

One of most common methods of saliva collection is the use of cotton swabs. In this situation, however, MLT levels are lower compared to passive saliva samples [43]. Polyester swabs can also be used to determine salivary MLT. Comparative studies of salivary collection using polymer swabs and passive saliva collection confirms the high correlation between the two methods [44].

## 5. Material Storage

Storage conditions of the material are an important factor of the analysis.

### 5.1. Blood

Blood is a common material for examinations. Therefore, blood collection procedures are well defined. Blood samples are collected into EDTA (ethylenediaminetetraacetic acid; versene acid, edetic acid) tubes and frozen at −20 ºC until further analysis, or they are taken directly into a tube where blood coagulates at 4 ºC. The next step is centrifugation, serum removal and freezing at −20 ºC. Graham et al. examined the stability of samples during 3-fold freeze–thaw cycles and did not observe the influence of this process on MLT level [25].

### 5.2. Urine

It was proven that a sample obtained to assess aMT6 level can be stored at room temperature no longer than five days without disturbing the stability of aMT6. The stability of aMT6 in urine and plasma is so high that it is possible to store samples for two years at −12 ºC and −20 ºC, respectively, with or without the use of preservatives [45]. 

### 5.3. Saliva

Saliva storage conditions depend on the site of material collection. Samples obtained in laboratory settings are often immediately frozen upon collection at −20 ºC. If the measurements are conducted by the subject, for instance at home, samples can be stored in refrigerators at 2–8 ºC and then collectively delivered to the laboratory.

## 6. Analysis

Currently, there are many available assays for the determination of MLT. Radioimmunoassay (RIA) and enzyme-linked immunosorbent assay (ELISA) are the most frequently used, both of which have several possible modifications. High performance liquid chromatography (HPLC) is less common whereas fast-scan cyclic voltammetry (FSCV) is a novel method on the market.

### 6.1. Radioimmunoassay (RIA)

RIA is widely used to determine MLT levels in saliva, blood and urine [2,25,46]. It allows quantitative assessment and is characterized by high sensitivity and specificity. The method is based on measurement of radioactivity of the radioisotope. Labeled MLT, anti-MLT antibodies and unlabeled MLT are used. Labeled MLT and unlabeled MLT compete for the antibody-binding site. With the increase in unlabeled MLT, the lower amount of labeled MLT forms complexes with antibodies. Then, unbound MLT is separated from the complexes, and the second antibody in the solid phase is used (Bulhmann RIA kit). The last stage is to measure radioactivity of the selected fraction. The higher the radioactivity of the sample, the lower the level of unlabeled MLT in the sample.

There are several assays on the market that have different antibodies or methods of separating the complexes from unbound MLT. The choice of the assay depends on the available resources, laboratory experience and sample specificity. For example, radioactive MLT 2-I125 iodomelatonin has an advantage over 3H-melatonin in low-volume or low-concentration melatonin samples [46]. Polyethylene glycol is used to separate MLT-free fractions from MLT-antibody complexes [47]. 

The comparison of ALPCO (American Laboratory Products Company, an importer and distributor of high quality research immunoassay kits) assay with Rollag assay and Elias USA assay showed similar results over a wide range of concentrations. Inter-assay variability was 11.3% for low standards and 5.4% for high standards [25].

### 6.2. Elisa

Elisa is an immunoenzymatic method used to determine MLT levels in saliva, blood and urine [46]. There are a number of modifications of ELISA, which have the common feature of the process of immobilizing the antigen on the solid phase and subsequently adding antigen-specific antibodies to the material. As a result, the antigen–antibody complex is formed. The medium is rinsed and the enzyme-labeled antibody is added. After adding the appropriate substrate, color reaction is achieved. Spectrophotometric measurement is the last stage of the process.

Ferrua and Masseyeff, in their study, showed a competitive immunoenzymatic method, i.e., enzyme immunoassay (EIA), using enzyme-labeled antibodies for the quantification of MLT in chloroform-extracted samples. The MLT level was measured in rat and human sera and human CSF. The sensitivity and specificity of the assay was comparable to RIA [48].

### 6.3. High Performance Liquid Chromatography (HPLC)

HPLC is a method characterized by high sensitivity and specificity. It allows detection of low concentrations of melatonin in small samples of biological material (20 microlitres). The speed of analysis is an additional advantage. This method is not suitable for the determination of melatonin in a sample containing a mixture of several indoles and does not provide information about the signaling kinetics [49].

It is a type of column chromatography. As a result of adding the solvent to the sample, a solution of known concentration and volume is obtained that is added to the top of the column. Mobile and stationary phases are used and the separation of the mixture is due to adsorption or dissolution processes, which depend on the stationary phase [50]. There are many variations of this method. 

Innuma F. et al. determined the level of melatonin in the rat pineal gland using HPLC with inverted phases. The derivations of melatonin were performed using hydrogen peroxide and sodium carbonate. Chromatography was carried out on the CO-96 column furnace. Linearity was established at 500 amols to 5 pmols and the sensitivity of the method presented was about 10 times higher than that of the previous methods [51]. Tomita et al. oxidized melatonin to N-[(6-methoxy-4-oxo-1,4-dihydroquinolin-3-yl) methyl] acetamide, obtaining a compound with higher fluorescence intensity. Linearity was established at 200 amol to 50 fmol, with a lower limit of quantification of 200 amol [52]. Hirano J. et al. proposed 5-methoxyindol-3-acetic acid (MIAA) as the internal standard because this compound has a strong fluorescence with a large displacement which is not disturbed by the change in mean pH. It is resistant to light and heat. It has been pointed out that the use of melatonin analogs as internal standards is beneficial and enables accurate measurements [53]. 

In their work, Khan SA et al. determined the levels of melatonin in saliva samples taken from children with sleep disorders using HPLC tandem mass spectrometers (MS/MS). Diluted saliva samples of 20 µl d7-melatonin, which is a deuterated internal standard, were used in the study. The test was carried out using the mobile phase A (0.1% aqueous formic acid) and B (15% methanol in acetonitrile containing 0.1% formic acid). Chromatography was performed on column C8. The study is a modification of the Errikson et al. study from 2003 [54]. The changes introduced concerned the abandonment of plastic products, the use of glass products previously subjected to careful cleaning and high-temperature evaporation for at least 48 h and the simplification of sample preparation. All the changes increased the measurement accuracy using a very small sample volume. Linearity was determined from 3.9 to 1000 pg/ml and measurement accuracy is 100%–105% (7.0–900 pg/ml). Studies on the use of this method in pharmacokinetic studies are in progress [55].

Liquid chromatography (LC)-MS with selected ion monitoring is a highly efficient and selective method. Motoyama A et al. determined the levels of endogenous salivary melatonin based on 1–1.5 ml salivary samples. The advantage of this method is fast analysis and minimal treatment of the biological fluid (adding internal standards and mixing). Chromatography uses a system of three columns in which sample purification, addition of an internal standard and separation are performed. D7-melatonin is used as an internal standard. Linearity was determined in the range of 5–250 and 100–2500 pg/Ml, with a measurement accuracy of 81%–108% and lower limit of detection 2.5 pg/ml [56].

### 6.4. Fast-Scan Cyclic Voltammetry (FSCV)

Fast-scan cyclic voltammetry (FSCV) is an innovative method used to determine MLT in the immune system.

Real-time measurements during an inflammatory disease can be a tool to provide information on the mechanism of immunomodulation. This method uses carbon fiber microelectrodes and is based on the measurement of the current in the system between the electrodes with the switching potential. The method is characterized by high sensitivity and it allows the observation of rapid changes in MLT level in the tissue due to the excellent temporal resolution. Hensley AL et al. simultaneously measured the level of melatonin, dopamine, serotonin, N-acetyl serotonin and histamine, and the results of the study proved the effectiveness of the method in selective measurement of substances, without disturbances resulting from a similar structure of chemical compounds. An additional advantage of the method is that the measurements are made without disturbing the structure of tissues. The use of a carbon electrode instead of diamond doped with boron decreased the melatonin detection limit 4 times. The detection limit was calculated at 24 ± 10 nM, while the slope of the calibration curve defines the sensitivity of the presented method and is 6608 nA/µM. Currently, it is used to study neurotransmission [49].

## 7. Factors Affecting the Measurement Results

### 7.1. Non-Modifiable Factors

#### 7.1.1. Genetic factors

It was found that the amount of secreted MLT depends on the volume of the active pineal tissue [57] that is genetically conditioned [58]. Kunz et al. noticed the relationship between the volume of the active pineal parenchyma and urinary excretion of aMT6 [59]. In addition, large inter-individual differences in the amplitude were observed with small intra-individual changes in urine aMT6 concentrations [60] and plasma MLT [13]. 

#### 7.1.2. Age

Many studies report a decrease in MLT secretion with age [61,62]. Higher MLT levels occur in children and adolescents compared to older subjects [63]. The change occurs between 20 and 30 years of age [64]. A significant MLT decline was observed in premenopausal women [65]. There can be many reasons for the decrease in MLT levels, such as calcification of the pineal gland with the progressive loss of pinealocytes or degenerative changes of suprachiasmatic nucleus neurons [66].

#### 7.1.3. Sex

Data on the variation in aMT6 concentration depending on sex are often contradictory. Some studies do not reveal any significant differences between women and men [26], whereas some reports indicate a clear relationship between MLT level and sex [17,67]. Differences in MLT synthesis due to naturally secreted sex hormones and hormonal contraceptive drugs [17] are considered to be the reason for the variation. The regulatory relationship between sex hormones and MLT is confirmed by the presence of MLT receptors in human reproductive organs, by the presence of sex hormone receptors in the pineal gland [68,69] and by animal studies [70]. 

### 7.2. Modifiable Factors

#### 7.2.1. Light

Light is the strongest stimulus that modifies the release of MLT [71]. Melanopsin, which is a photoreceptor discovered in 2000, is a visual pigment in intrinsically photosensitive retinal ganglion cells (ipRGC receptors). The greatest MLT suppression occurs when ipRGCs absorb a quantum of light at a short wavelength (mainly 460–480 nm). For the human eye it is blue–cyan color [72,73]. Electromagnetic waves of such lengths are present in the solar radiation band and in the radiation emitted from artificial light sources (e.g., devices with LCD screens and LED lighting). Melanopsin shows the highest sensitivity to light at a wavelength of approximately 482 nm [1] and then melanopsin deactivation occurs.

There are also practical (not always successful) attempts to modify the evening lighting with the exclusion of short waves [73]. Many studies relate to the influence of laptops and smart phones in the evening on MLT levels [74]. Among users of tablets and devices with short-wavelength light emission, it was observed that working on an electronic document suppresses MLT levels and delays the circadian rhythm compared to users of printed material [74]. Because exposure to light in the evening and at night disturbs the circadian rhythm, causing inhibition of MLT release and a phase delay in the circadian rhythm [75,76], research on appropriate regulations seems to be important. In turn, high-frequency light flickering has little effect on MLT secretion [77].

#### 7.2.2. Seasonal Changes

MLT secretion rhythms in various seasons are statistically significantly different [78]. Of note, seasonal changes in day length (photoperiod) modify the duration of nocturnal MLT secretion in many vertebrates, including humans [79]. Lengthening of the dark phase that is observed in natural conditions in the winter results in lengthening of the nocturnal secretion of this hormone. 

#### 7.2.3. Posture During the Examination

Data related to the impact of the posture on variations in MLT level are not conclusive. Deacon et al. measured MLT in saliva and plasma and indicated an increase in MLT concentration in both body fluids when an upright position was adopted after lying recumbent and a decrease in concentration after reversing the postural position [20]. However, these observations were not confirmed in other studies [36].

#### 7.2.4. Physical Activity

Physical activity can modify MLT secretion [80]. Exercising at night delays the onset of MLT release [71,81] and may even lead to the inhibition of MLT secretion [82]. These observations are related to situations when physical exercise precedes the phase of MLT increase. However, when MLT level is high during training, a further increase is observed [83]. Buxton et al. reported a significant increase in MLT among some of the subjects who performed physical exercise in the evening [80]. Marrin et al. investigated salivary MLT level and found an elevation in MLT levels after physical exercise in their subjects. These changes also depended on the time of the day when the training was performed [83].

## 8. Comorbidities

Melatonin is currently a subject of many clinical trials for its possible future use as a marker for many diseases. In our analysis, we focused only on such conditions that may be related to MLT metabolism or the method of sample collection.

### 8.1. Ophthalmic Diseases

The pathway of the light stimulus starts from retinal ganglion cells. Therefore, ophthalmic diseases will affect MLT secretion. For instance, the analysis of sleep patterns in blind individuals shows a high incidence of sleep disorders. In individuals without conscious light perception, sleep disorders are more frequent and more intense compared to persons with a preserved low degree of light perception [84]. The majority of completely blind individuals experience continuous circadian dyssynchrony due to the failure related to the transportation of light information to the hypothalamic circadian clock. It results in cyclic episodes of poor sleep and dysfunction during the day [85]. However, a small group of such patients has normal MLT secretion. This indicates that rod-cone disorders may coexist with normal retinal ganglion cell function [3].

### 8.2. Spinal Cord Injuries

Spinal cord injuries affect MLT secretion. The anatomical location of spinal cord injury is significant. The comparison of a control group and patients with cervical and thoracic injuries showed that only in patients with thoracic injury did the MLT secretion pattern correspond to that in the control group [86,87]. Other results in the group with cervical spine injuries were related to the fact that the pathway of pineal gland stimulation with the use of light runs through the spinal cord in the cervical segment. Injury to this segment was related to the fact that the nocturnal increase in MLT level did not occur.

### 8.3. Liver and Kidney Diseases

The liver is the main organ metabolizing MLT. Therefore, its failure significantly affects the level of this hormone. Laboratory studies showed a delay in the increase in plasma MLT and a persistently elevated level of MLT during the day. In cirrhotic patients, the half-life of this hormone was prolonged throughout the day to approximately 100 min [88].

Kidney diseases can also affect MLT secretion. Several factors have an impact on the production of MLT in patients with chronic kidney disease (CKD). First, in renal failure, a decrease in MLT levels may occur as a result of impaired beta-adrenergic receptor-mediated response [89]. Secondly, metabolic acidosis and a decrease in muscle tone in the airways due to accumulation of urea toxins result in an increased incidence of sleep apnea in CKD, which is associated with increased MLT levels in the afternoon [90]. Thirdly, uremia, which is associated with daytime sleepiness, causes disorders of the sleep–wake cycle [91]. The secretion pattern of MLT can also be dysregulated due to erythropoietin-deficiency anemia, which is often found in patients with CKD [92]. Finally, drugs such as beta-blockers and benzodiazepines can impair MLT production and therefore a nocturnal peak is observed [93]. A decrease in urinary aMT6 levels has also been observed [94].

### 8.4. Periodontal Disease

Collecting saliva directly from the oral cavity requires determination of the influence of periodontal disease on the obtained results. Two independent studies showed decreased MLT levels in saliva and in the gingival fluid in patients with inflammation compared to a healthy control group [95,96]. Balaji et al. also confirmed lower MLT concentration in saliva and a slight increase in plasma levels in patients with chronic periodontitis. However, these differences were not statistically significant. In turn, a significant decrease in MLT level in the gum tissues was observed in patients with chronic inflammation [97]. Experiments on rats with induced periodontitis showed decreased plasma MLT levels [98].

## 9. Medications

Some medications do not affect the level of MLT whereas some drugs significantly change it.

Drugs that decrease MLT concentration include the following: β1-adrenergic blockers [13,99,100,101,102], α2 adrenergic agonist [103] and benzodiazepines used regularly in high doses [27,104]. Triazolam, which is a very short-acting benzodiazepine derivative, does not affect plasma MLT levels [105]. Increased MLT concentration has been observed in patients who regularly receive antidepressants and MAO inhibitors [13]. 

The scientific opinion on the influence of antiepileptic drugs on melatonin levels is ambiguous. In their work, Gupta M et al. noted higher levels of melatonin in a group of patients using carbamazepine compared to patients taking valproic acid [106]. On the other hand, the studies by Praninskie et al. investigating melatonin levels in saliva [107], and those of Dabak O et al. investigating plasma melatonin levels [108] in pediatric patients, do not confirm these observations. The effects of drugs are discussed in Table 1.

The second group of drugs with uncertain influence on melatonin secretion are oral contraceptives. Some authors did not observe the effect of hormone replacement therapy on aMT6 concentration in morning urine samples [33], whereas others indicated a potential modification of MLT production after the use of oral contraceptives with ethinylestradiol [17] or increased nocturnal MLT secretion in women on this medication [114,115]. The effects of sex hormones are discussed in Table 2. 

## 10. Conclusions

Basic requirements for correct determination of MLT levels and the validity of the comparative results of such studies are related to the standardization regarding conditions of storage and place of performing the assays.

None of the assay methods for MLT are superior and the choice of the appropriate method depends on many factors [116], one of them being the aim of the examination.

The lack of rigid guidelines for sampling and analysis prompted the researchers participating in the 2005 Associated Professional Sleep Societies to try to reach a consensus on this topic. The paper uses a modified RAND process based on voting. The result of the cooperation was a draft of developed recommendations supported by all researchers. The main sampling guidelines are contained in Table 3.

The divergences resulting from the application of different DLMO definitions are proposed to be solved by using, in the publication, in addition to the previously selected threshold, an additional low DLMO threshold (e.g., for saliva <3 pg/mL or for plasma 2 SD or <10 pg/mL) [116]. 

DLMO is best obtained by serum/plasma MLT measurements, followed by salivary assay, due to the fact that blood and saliva samples show MLT concentrations at the exact time of sample collection, while urine measurements show the accumulated amount of 6-sulfatoxymelatonin during an interval of time [39,41]. Furthermore, saliva and serum MLT concentrations undergo almost simultaneous changes, as shown by exogenous MLT administration [35].

The frequency of sampling is another important factor determining the quality of the obtained results. The more frequent the sampling, the more accurate the results are. Properly determined secretion profiles of MLT can be mathematically modeled, which in turn facilitates the comparison of the parameters characterizing the secretion cycles, including DLMO [14,15,16].

The method of sample collection is another important aspect. It can be non-invasive or invasive and each approach has its advantages and disadvantages. Due to non-invasive collection and no requirement for technical skills related to sampling, patients themselves can obtain saliva or urine samples, which allows the use of these methods in epidemiological and pediatric studies. When saliva is sampled, the need to stimulate the patient during the material collection can be the disadvantage at night hours. The least invasive method of probe collection to assess nocturnal MLT production is the measurement of aMT6 from the morning urine sample, without using an intravesical catheter. There is no risk of transmission of infection from patients to health care personnel, which is the advantage of non-invasive methods. Another advantage of urinary and salivary MLT measurement is the lack of special requirements related to the storage conditions of the samples. Subjects may collectively deliver all samples to the laboratory after sample collection. The assessment of the circadian phase based on saliva samples collected under laboratory conditions is significantly correlated with the results obtained from home sampling [95].

Measurement of plasma/serum MLT levels is the basic invasive method. It is possible to collect many samples without sleep disturbance despite the invasiveness associated with sample collection.

The measurement method should be adapted to the subject due to the possible occurrence of difficulties in the collection of blood, urine or saliva. If the subject has low MLT levels, determination in saliva and urine can be difficult. In such cases, plasma MLT measurement should be the method of choice. In critically ill patients with multi-organ failure, the correlation between aMT6 levels in urine and plasma may be impaired [9]. In such conditions blood sampling will be the preferred method. Financial issues, budget constraints in research centers, the availability of the equipment and personnel as well as the possibility of utilization of radioactive waste are also of crucial importance.

## Figures and Tables

**Figure 1 ijerph-17-01916-f001:**
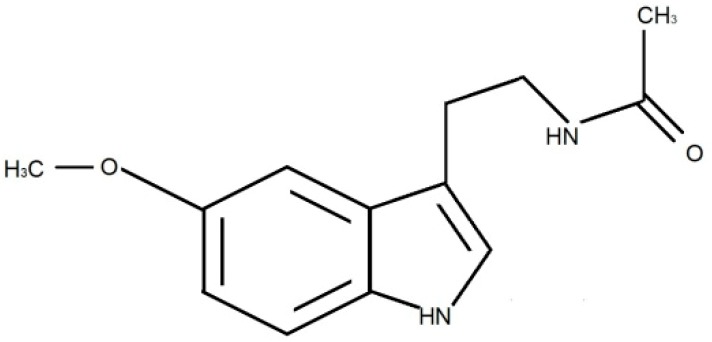
Chemical structure of N-acetyl-methoxytryptamine.

**Figure 2 ijerph-17-01916-f002:**
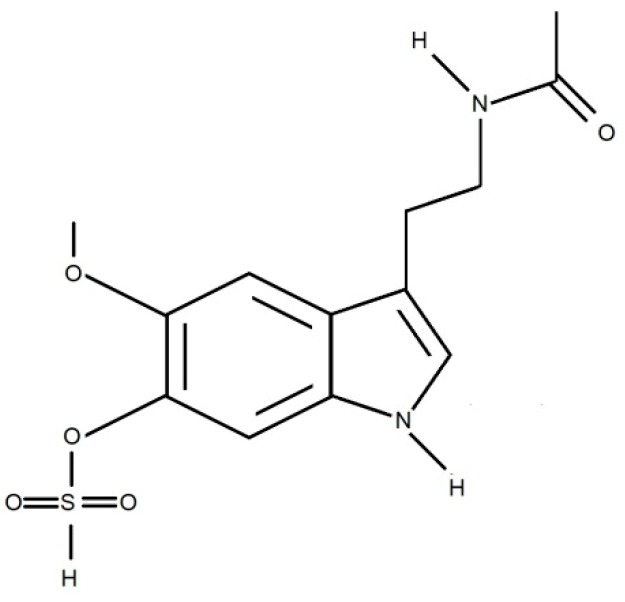
Chemical structure of 6-sulfatoxymelatonin.

**Figure 3 ijerph-17-01916-f003:**
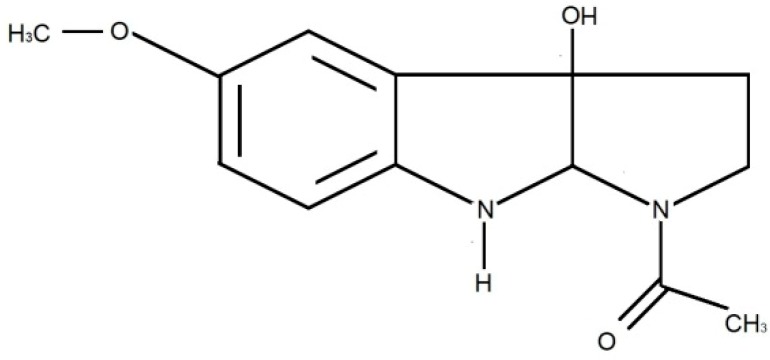
Chemical structure of hydroxymelatonin.

**Table 1 ijerph-17-01916-t001:** Effects of drugs on melatonin secretion.

References	Drug	Effects on Melatonin Secretion
Cowen PJ et al., 1985 [99]	beta-blockers (propranolol, atenolol)	lower mean melatonin concentration
Brismar K et al., 1987 [100]	beta-blockers	decreased night-time melatonin secretion
Rommel T et al., 1994 [101]	beta-blockers (propranolol, ridazolol)	decreased melatonin secretion
Stoschitzky K et al., 1999 [102]	beta-blockers (atenolol)	decreased melatonin secretion
Takaesu Y et al., 2015 [109]	beta-blockers	no significant differences
Muñóz-Hoyos A et al., 2000 [103]	α2 adrenergic agonist (clonidine)	decreased melatonin secretion
Murphy PJ et al., 1996 [110]	nonsteroidal anti-inflammatory drug (NSAIDs) (aspirin or ibuprofen)	decreased nightly melatonin secretion
Monteleone P et al., 1989 [111]	GABAergic drug (diazepam)	decreased night-time melatonin secretion
Monteleone P et al., 1997 [112]	GABAergic drug (sodium valproate)	decreased melatonin secretion
Gupta M et al., 2006 [106]	carbamazepine+melatonin (CBZ+MLT) valproate+melatonin (VPA+MLT)	melatonin levels in patients receiving CBZ+MLT were higher than those of the VPA+MLT recipient group The observed difference in melatonin levels could be attributed to the difference in antiepileptic drugs
Praninskiene R et al., 2012 [107]	antiepilepticdrugs	no significant differences
Dabak O et al., 2015 [108]	antiepilepticdrugs	no significant differences
McIntyre IM et al., 1988 [104]	benzodiazepines (alprazolam )	decreased night-time melatonin secretion
Copinschi G et al., 1990 [105]	short-acting benzodiazepine (triazolam)	no significant differences
Claustrat B et al., 2005 [13]	monoamine oxidase inhibitors (MAO)	increased melatonin secretion
Claustrat B et al., 2005 [13]	tricyclic antidepressants	increased melatonin secretion
Skene DJ et al., 1994 [113]	the specific serotonin uptake inhibitor (fluvoxamine)	increased nocturnal plasma melatonin concentrations
Skene DJ et al., 1994 [113]	the noradrenaline uptake inhibitor (desipramine)	increased evening plasma melatonin concentrations

**Table 2 ijerph-17-01916-t002:** Effects of sexual hormones on melatonin secretion.

References	Participants	Effects on Melatonin Secretion
Kostoglou-Athanassiou I et al., 1998 [114]	- women on oral contraceptives - women not on oral contraceptives	overall melatonin secretion was augmented
Cook MR et al., 2000 [33]	- women who took hormone replacement therapy - women who did not take hormone replacement therapy	no effect observed
Burgess HJ et al., 2008 [115]	- females who used hormonal birth control therapy - females who did not use hormonal birth control therapy - males	longer duration time of melatonin secretion
Gunn PJ et al., 2016 [17]	- women on oral contraceptives - males	significantly elevated plasma melatonin levels in women, no significant differences in aMT6 levels

**Table 3 ijerph-17-01916-t003:** Sampling guidelines for melatonin according to Benloucif S et al. 2011.

Material	Determined Substances	Sampling Periods	Lighting	Body Posture	Basis for Evaluating the Circadian Rhythm Phase
urine	aMT6s	every 2 to 8 h for 24 to 48 h or the first morning urine sample	not applicable	restriction of motor activity and change of body position before and during sampling	the timing of the acrophase
saliva	melatonin	every 30 to 60 min starting at least one hour before and throughout the expected increase in melatonin levels	<30 lux		DLMO
blood	melatonin	frequent sampling with a catheter inserted at least 2 h before the expected increase in melatonin levels	<30 lux		DLMO

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
