# Peer review of "Melatonin-Measurement Methods and the Factors Modifying the Results. A Systematic Review of the Literature"

_ijerph, 2020, doi:10.3390/ijerph17061916_

Round 1

Reviewer 1 Report

This manuscript reports a state-of-the-art review about the measurement methods of melatonin levels and modifiable and unmodifiable factors that can affect the results. The literature covered is updated.

The manuscript could be accepted after minor revisions as followings.

  • A figure with the chemical structure of MLT, 6-hydroxymelatonin, 6-sulfatoxymelatonin and 3-hydroxymelatonin should be included at the end of the introduction.
  • The style of the references should be checked according to the guidelines of the journal. Sometimes the author’s name is underlined and sometimes is underlined the name of the journal or the title.
  • On page 1, line 35, “after its secretion is immediately diffuses …” instead of “after its secretion is immediately diffuses ...”.
  • On page 5, line 196, the meaning of ALPCO should be defined in the main text

Author Response

In response to Reviewer 1 comments, the following changes have been made:

-Drawings showing structural patterns of the detected compounds were prepared

- The style of the references has been improved in accordance with the guidelines. Due to the modification of the manuscript, the bibliography from number 50 inclusive has been changed.

- Line 36, "after its secretion is immediately diffuses ..." - the sentence was corrected

- Lines 206-207 ALPCO abbreviation was developed  

Reviewer 2 Report

Authors have attempted to present a review of melatonin measurement methods. These exist diversity of methods however gold standards do exist. Throughout this report authors have not highlighted gold standards or widely accepted standards.

section 8. medications -Line 338-346 is underdeveloped - a table with drugs and their effects on melatonin measurements would be helpful.

Line 211-  HPLC needs a major rewrite - currently the authors have attempted an explanation of the HPLC as a tool but no meaning description of the usefulness of the tool to measure MLT from biosamples. especially analytical parameters, detectors, instrumentation limitations etc

LC MS/MSn has amplified the utility of HPLC.

Similarly Lines 218-223 FSCV is not sufficiently described. "High sensitivity" "low sensitivity" - what are the detection limits???

Line - 352 "None of the assay methods for MLT is superior" - a sweeping statement -undermining some protocols that are widely regarded. Authors should make an attempt to carefully word this and other blanket/sweeping statements

words or sentences repeated as in

- Line 69-70 No uniform DLMO estimation protocol has yet been created. No uniform DLMO estimation protocol
 has yet been created. 

Author Response

In response to Reviewer 2 comments, the following changes have been made:

- Lines 423-433, Manuscript has been supplemented by the publication of Benloucif S et al. from 2011, which is a set of generally accepted sampling rules for melatonin.  This publication aims to simplify the planning of the study for Clinicians and stresses that there is no single universal scheme and that the choice of material and method depends on the population under study. In addition, the paragraph referring to DLMO, lines 65-100, highlights the most commonly used sampling protocols

- Lines 395-416 - The paragraph on the influence of drugs on melatonin secretion has been extensively expanded and a table summarizing numerous studies has been created.

- Lines 222-261 - The paragraph concerning the method (HPLC) has been extended, attention has been paid to the available modifications of the method and the resulting differences in sensitivity and precision of melatonin determination.

- Lines 262-276 In the paragraph on the method (FSCV) - attention was paid to the possibility of using the method for simultaneous determination of melatonin and structurally similar compounds. High sensitivity is presented in numerical form.

- Line 78 – The repeating sentence was deleted.

Reviewer 3 Report

The review not up the the standards of International Journal of Environmental Research and Public Health journal.

  1. There is no proper methodology and narration of the review.
  2. Poor interpretation.
  3. no proper citations.
  4. no proper journal format including reference format.
  5. very elementary writing.. no diagrams..no tables..

I cant comment on more this manuscript. its unworthy to read, i feel this is not up to the mark to publish in IJERPH Journal. 

Author Response

 We have changed our manuscript following reviewer's comments.

Round 2

Reviewer 2 Report

Authors revisions are appropriate. 

The review has enhanced significantly and recommend  to accept for publication. 

Reviewer 3 Report

The Authors provided substantial information in the revised version.

Authors revised all concerns raised by the reviewers and this must be accepted.

The current revised version is worth to read and accepted.